# PREP: Pre-inference Guided Token Pruning for Efficient Vision-Language Models

## Abstract

Recent Visual-Language Models (VLMs) have demonstrated strong fine-grained perception capabilities across a wide range of Visual Question Answering (VQA) tasks. However, this advantage comes at the cost of a rapidly increasing number of visual tokens, leading to substantial computational and memory overhead. Existing training-free methods adopt fixed-layer or layer-by-layer pruning, which disrupts modality fusion before alignment and leads to significant performance degradation under high pruning ratios. In this study, we observe that after the early stage of modal fusion, cross-modal attention not only accurately identifies regions of interest but also demonstrates less sensitive to pruning. Building on this, we propose **PREP**, a training-free method that identifies optimal pruning layer via patch-level pre-inference, thereby avoiding the loss of fine-grained details under stepwise pruning. Specifically, PREP identifies the the layer with accurate cross-modal alignment using an **E**ntropy–**KL** divergence (EKL) score derived from the Information Bottleneck principle, and then retains tokens at this layer that are critical for visual integrity and semantic alignment during full inference. Experiments on LLaVA-1.5-7B show that with only **9** visual tokens and half of the layers used in pre-inference, PREP preserves **96.2%** of the original performance while retaining just **16** visual tokens (**3%**), leading to a **67%** reduction in KV-cache usage and a **1.66×** acceleration in inference speed. We have presented our code in the supplementary materials.

## 1 Introduction

Visual-Language Models (VLMs) have advanced rapidly in recent years (e.g., LLaVA-1.5 Liu et al. (2023), InternVL3 Lu et al. (2025), GPT-4o Hurst et al. (2024)), pushing the frontier of multimodal reasoning and fine-grained perception. For instance, LLaVA-1.5 encodes each image into a fixed 576 visual tokens, already far exceeding the number of textual tokens and straining LLM context capacity. More recent models such as InternVL3 adopt substantially larger visual encoders, producing over 6000 tokens per image to capture fine-grained details. While such designs greatly enhance perception, it also introduces substantial computational and memory overhead, thereby limiting the scalability and real-time deployment of VLMs.

Existing token compression strategies fall into training and training-free methods. Training methods redesign the encoder or LLM architecture to inherently reduce visual token overhead. For example, PDrop Xing et al. (2024) trains models to adapt to pruned token inputs by progressively dropping tokens during training , while LLaVA-Mini Zhang et al. (2025b) introduces a lightweight cross-attention module before LLM and reduce into one visual token. Although effective, these approaches require substantial retraining and often lack portability across different VLM backbones. In contrast, training-free methods directly prune tokens at inference without retraining. Representative approaches such as SparseVLM Zhang et al. (2024b), TopV Yang et al. (2025a), and Dymu Wang et al. (2025) dynamically prune tokens layer by layer based on cross-modal attention, while others like Minimonkey Huang et al. (2024) and VScan Zhang et al. (2025a) select a fixed layer to prune. However, both of them fail to preserve performance under high visual token pruning ratios (e.g., more than 90%), which we attribute to their neglect of the distinct functional roles of different layers, causing them both to miss *when* textual and visual information become aligned and discard local details during pruning. As shown in Fig. 1, in the early layers, LLaVA-1.5-7B remains in the stage of visual–textual fusion, where similarity between prompt and image tokens is broadly

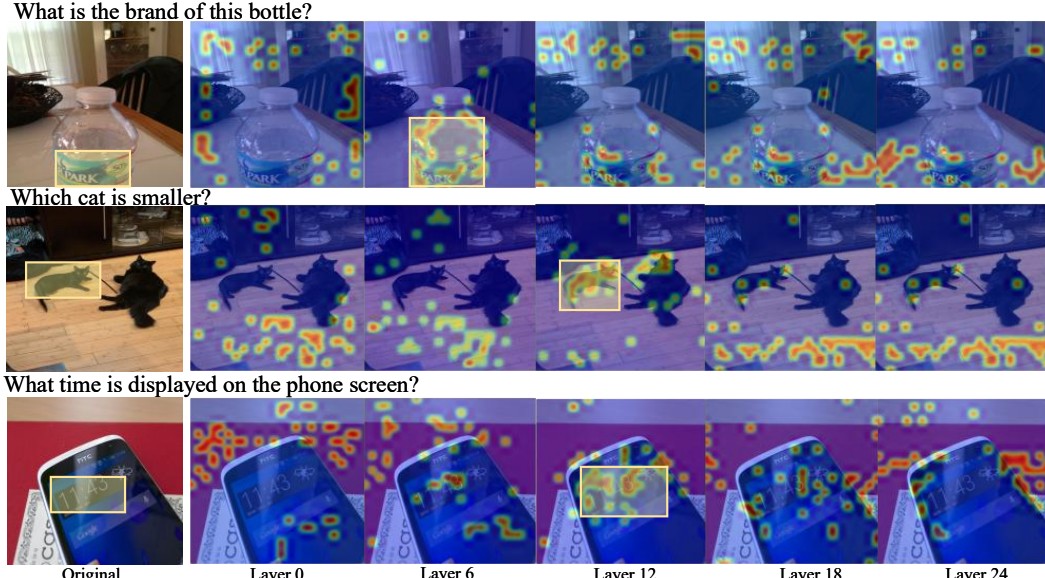

Figure 1: Attention matrices of LLaVA-1.5-7B across different layers, after filtering out tokens with text-image similarity below 70% of the maximum. Yellow boxes indicate regions of interest.

distributed and fails to capture the regions of interest (ROI). In the middle layers, cross-modal alignment emerges, yielding accurate localization of ROI. In the late layers, the models exploit high-level semantic representations for task-specific reasoning, while attention becomes dispersed once again. In Fig. 2, this trend is further confirmed by our observation that pruning in the middle and late layers incurs significantly less performance degradation than in the early layers. Based on this finding, we argue that pruning should be performed as soon as the layers completing modal fusion are identified. This not only ensures efficiency but also mitigates the loss of fine-grained information that typically occurs within layer-wise or fixed-layer pruning strategies.

Building upon this insight, we introduce a **PRE**-inference guided **P**runing strategy, termed **PREP**. Firstly, PREP averages a fixed number of visual tokens and get patch-level visual tokens as a cheap proxy for observing cross-modal alignment. During pre-inference, PREP computes a visual important distribution from cross-modal similarity of each layer and then identifies the optimal pruning layer with the maximize **E**ntropy and **KL**-divergence score(EKL), which is derived from information bottleneck principle and signals accurate modal-alignment. Finally, at the selected layer, PREP retains visual tokens according to multi-modal importance scores computed by combining visual–visual and visual–prompt attention matrices, thereby preserving tokens critical for both visual integrity and semantic alignment.

Our experiments on 9 VQA benchmarks demonstrate that PREP retains **96.2%** of the original performance even with an **97%** reduction in visual tokens. Meanwhile, KV cache usage is reduced by **67%**, and inference is accelerated by **1.66×**, leading to substantial reductions in latency and improved memory efficiency. These results highlight our method ability to significantly compress visual tokens while preserving performance on challenging fine-grained vision-language tasks.

## 2 RELATED WORK

Recent advancements in Vision-Language Models (VLMs) focus on improving efficiency through visual token compression. A promising and widely explored direction centers on trainable compression techniques. Key examples of such trainable approaches include: LLaVA-Mini Zhang et al. (2025b) reduces the number of vision tokens by using a query-based compression module. Similarly, Vision Concept Models (VCM) Luo et al. (2025) dynamically extract the most relevant visual concepts based on task-specific instructions, optimizing the model's performance. The Progressive Visual Token Compression(PVC) Yang et al. (2025b) method also enhances efficiency by focusing on key visual features by introducing Progressive Visual Token Compression module, while PDrop Xing et al. (2024) introduces a dropout mechanism across a pyramid structure in the visual encoder, improving feature se-

lection. These methods aim to streamline visual processing while maintaining or enhancing model performance. However, they often require retraining for each specific model, leading to significant resource consumption and limiting their scalability in diverse applications.

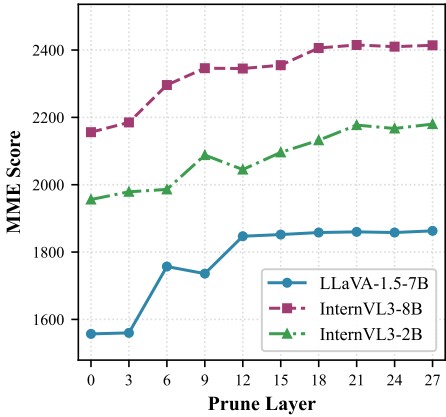

Figure 2: Performance on the MME Zhang et al. (2024a) when pruning 85% of tokens at different layers.

Training-free methods compress or select visual tokens layer-by-layer during the pre-processing phase. SparseVLM Zhang et al. (2024b) introduces a rank-based strategy to adaptively determine sparsification ratios and uses token recycling to compress pruned tokens. HiRED Arif et al. (2025) employs a token-dropping method within a fixed token budget, allocating tokens based on the attention of the CLS token in ViTs. TopV Yang et al. (2025a) formulate token pruning as a layer-wise optimization problem, accurately identifying important visual tokens. Dymu Wang et al. (2025) reduces token embeddings through Dynamic Token Merging (DToMe) and simulates full-token sequences with Virtual Token Unmerging (VTU) to maintain performance without fine-tuning. Minimonkey Huang et al. (2024) directly prunes tokens according to the cross-attention of the second layer, while VS-

can Zhang et al. (2025a) prunes at the 16 layers. While these approaches avoid retraining, their layer-wise or fixed-layer compression fails to identify the modality-alignment layers, thereby discarding critical ROI regions and undermining fine-grained perception, ultimately leading to performance degradation.

# 3 METHOD

In this section, we introduce our token pruning framework for VLMs. We begin by analyzing cross-modal alignment from information bottleneck principle. Building on this insight, we present Entropy and KL-divergence based Layer score (EKL) for layer selection during pre-inference. Then, we introduce multi-modal token score for token pruning during full-inference. The overall framework is shown in Fig. 3.

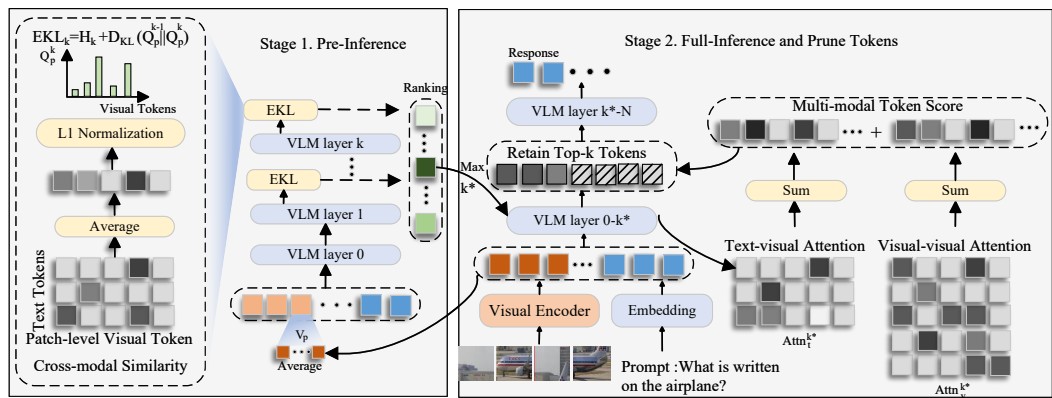

Figure 3: Overview of PREP framework. Stage 1: PREP identifies pruning-friendly layer with patch-level pre-inference tokens via EKL score. Stage 2: PREP combines visual–visual and text–visual attention to retain the most informative tokens.

## 3.1 PRELIMINARY ANALYSIS

VLMs generate textual responses conditioned on images and prompts. An image input $\mathbf{I} \in \mathbf{R}^{W \times H \times 3}$ is first encoded by a transformer-based visual encoder (e.g., ViT Dosovitskiy et al. (2020)) and then projected via an MLP to the required feature dimension $D$, yielding visual tokens $\mathbf{V} \in \mathbf{R}^{N \times D}$, where $N$ is the number of tokens. Meanwhile, the text prompt is embedded through

the embedding layer as $\mathbf{T} \in \mathbf{R}^{M \times D}$, where $M$ denotes the prompt length. Previous pruning methods (Zhang et al., 2024b; Wang et al., 2025) typically compute cross-modal similarity between $\mathbf{V}^k$ and $\mathbf{T}^k$ or rely on attention scores $\mathbf{Attn}^k$ from the $k$-th layer to determine the number of tokens to prune. However, they fail to identify the precise layer where cross-modal alignment emerges, leading to the loss of fine-grained information. Specifically, as shown in Fig. 1, in some layers, the image tokens in the prompt-related regions exhibit high similarity, showing explicitly modal alignment.

Based on this observation, we first introduce $\mathbf{Q}^k$ to reflect the alignment result between text and vision at the $k$-th layer, which can be computed it as:

$$\mathbf{Q}^k = \frac{\text{Mean}_j\left[\text{Softmax}\left(\frac{\mathbf{V}^k(\mathbf{T}^k)^\top}{\sqrt{D}}\right)\right]}{\sum_i \text{Mean}_j\left[\text{Softmax}\left(\frac{\mathbf{V}^k(\mathbf{T}^k)^\top}{\sqrt{D}}\right)\right]_i}, \quad \mathbf{Q}^k \in \mathbf{R}^N, \tag{1}$$

where $\text{Mean}_j[\cdot]$ denotes averaging over the text-token dimension $j$, and the summation index $i$ corresponds to the visual dimension, corresponding to the average and L1 normalization in Fig. 3. In the encoding results of this layer, visual tokens with higher similarity to the prompt will have a higher $\mathbf{Q}^k$ value, while it is ensured that $\mathbf{Q}^k$ follows a probability distribution. Then, we introduce the target distribution $Y$ as the underlying visual importance, corresponding to prompt-relevant regions.

To evaluate whether the visual tokens of the current layer are aligned with the prompt and faithfully reflect the relevant regions, $\mathbf{Q}^k$ should simultaneously (i) preserve information about $Y$, ensuring faithful identification of semantically relevant tokens(higher $I(\mathbf{Q}^k; Y)$), and (ii) remain maximally compressed relative to the previous layer $\mathbf{Q}^{k-1}$(lower $I(\mathbf{Q}^k; \mathbf{Q}^{k-1})$), thereby discarding redundant information. This trade-off is consistent with the objective of the Information Bottleneck (IB) theory and can be expressed by the following objective:

$$\mathcal{L}_{IB} = I(\mathbf{Q}^k; Y) - \beta I(\mathbf{Q}^k; \mathbf{Q}^{k-1}), \tag{2}$$

where $I(\cdot; \cdot)$ denotes mutual information and $\beta > 0$ is a balancing parameter. As mentioned above, a larger value of $\mathcal{L}_{IB}$ indicates a higher cross-modal alignment quality for this layer. We then expand this target as:

$$I(\mathbf{Q}^k; Y) - \beta I(\mathbf{Q}^k; \mathbf{Q}^{k-1}) = (1 - \beta)H(\mathbf{Q}^k) - H(\mathbf{Q}^k \mid Y) + \beta H(\mathbf{Q}^k \mid \mathbf{Q}^{k-1})), \tag{3}$$

where $H(\cdot)$ denotes entropy and $H(\cdot \mid \cdot)$ conditional entropy. However, directly computing the conditional entropy in Eq. 3 is intractable: the ground-truth target distribution $Y$ is inaccessible during inference, and the visual attention distribution from $\mathbf{Q}^{k-1}$ to $\mathbf{Q}^k$ involves complex transformer internal computations. To resolve this, we next propose a feasible approximation to the IB objective using **E**ntropy and **KL**–divergence(EKL) score.

## 3.2 Entropy and KL–divergence Score(EKL)

As mentioned above, $H(\mathbf{Q}^k \mid Y)$ quantifies the uncertainty of $\mathbf{Q}^k$ when the underlying visual importance $Y$ is known. Intuitively, if $\mathbf{Q}^k$ deviates significantly from $Y$(e.g., the attention of model focuses on non-ROI regions), the uncertainty of $\mathbf{Q}^k$ cannot be effectively reduced even with prior knowledge of $Y$—this implies a larger $H(\mathbf{Q}^k \mid Y)$. In addition, according to our previous observations, obvious modal-alignment appears after early modal-fusion layers, indicating a small and approximately constant $H(\mathbf{Q}^k \mid Y)$ for the middle layers. To identify this range, we calculate, for each layer of LLaVA-1.5-7B, the **ratio of** the intersection area between the top 75% attention-weighted areas predicted by $\mathbf{Q}^k$ and the ROI **to** the area of the ROI, which is termed as intersection over ROI (IoR) and described in detail in Fig. 4.

If the conditional entropy $H(\mathbf{Q}^k \mid Y)$ is small, this means that most of the regions attended to by $\mathbf{Q}^k$ can be predicted when $Y$ is known; in this case, the intersection between these predicted regions and the ROI will be larger, corresponding to a higher IoR. In Fig. 4, the IoR values remain consistently high with minimal fluctuations across layers 6–15. This stable alignment between $\mathbf{Q}^k$ and $Y$ implies that the conditional entropy $H(\mathbf{Q}^k \mid Y)$ remains relatively constant. In Appendix A.1, we observe a similar prunable range in both InternVL and QwenVL. This phenomenon may arise from the fact that current VLMs are pre-trained primarily using the next-token prediction objective and share largely unified LLM-style architectures, which in turn leads to similar layer-wise attention

patterns. Similar to the patterns observed in Fig. 1, the shallower layers primarily facilitate cross-modal fusion, whereas the deeper layers progressively transition toward task-specific reasoning. Consequently, the degree of alignment between $\mathbf{Q}^k$ and $Y$ exhibits substantially larger fluctuations in these regions, suggesting that the conditional entropy $H(\mathbf{Q}^k \mid Y)$ cannot be approximated as invariant across these layers.

Similarly, for the second term in Eq. 2, we approximate $H(\mathbf{Q}^k \mid \mathbf{Q}^{k-1})$ by measuring the divergence between the attention distributions of consecutive layers. Intuitively, if $\mathbf{Q}^k$ carries little new information beyond $\mathbf{Q}^{k-1}$, the two distributions will be highly similar, resulting in a small conditional entropy. Conversely, a large divergence indicates that $\mathbf{Q}^k$ introduces substantial novel information relative to $\mathbf{Q}^{k-1}$. Following this intuition, we compute the KL divergence $D_{\mathrm{KL}}(\mathbf{Q}^k \| \mathbf{Q}^{k-1})$ at each layer as a practical surrogate for $H(\mathbf{Q}^k \mid \mathbf{Q}^{k-1})$. Accordingly, we define the EKL score for layer $k$:

$$\mathrm{EKL}_k = \mathcal{H}(\mathbf{Q}^k) + D_{\mathrm{KL}}(\mathbf{Q}^k \| \mathbf{Q}^{k-1}). \tag{4}$$

We also provide a detailed proof using information-theoretical bounds in Appendix A.2, proving that EKL can be used as a computable lower bound for $\mathcal{L}_{IB}$. Based on the above analysis, for the selected layers where $H(\mathbf{Q}^k \mid Y)$ remains approximately constant, a larger $\mathrm{EKL}_k$ implies that the value of the remaining term in Eq. 3 is larger, which in turn indicates a higher degree of cross-modal alignment for this layer.

However, directly computing the EKL score at the token level during pre-inference would be computationally intensive. For efficiency, we partition $\mathbf{V}$ into r groups $V_r \in \mathbf{R}^{r \times L \times D}$ and average over the first dimension ($r$) to obtain patch-level tokens $\mathbf{V}_p$:

$$\mathbf{V_P} = \frac{1}{L} \sum_{k=1}^{L} \mathbf{V_r}[:, k, :], \quad \mathbf{V}_p \in \mathbf{R}^{r \times D}, \tag{5}$$

where the averaging operation aggregates pixel-level features within each patch to preserve patch-wise semantics.

To validate its feasibility for pre-inference, we obtain the IoR of patch-level distributions $\mathbf{Q}_p^k$ with the same setting as token-level IoR. As illustrated in Fig. 4, the high-attention regions remain well aligned across both representations in the middle layers.

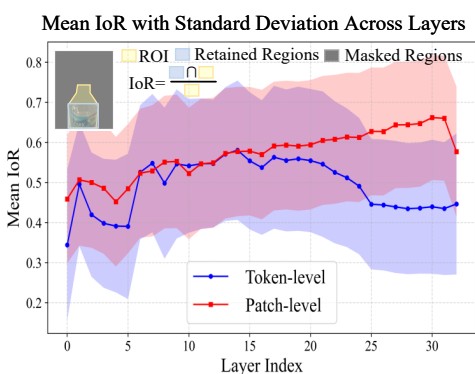

Figure 4: IoR means the intersection area between the top 75% attention-weighted areas predicted by $\mathbf{Q}^k$ and the ROI over the area of the ROI on VizWiz Chen et al. (2022).

These findings suggest that patch-level encoding faithfully preserves the critical semantics captured by token-level encoding, thereby enabling reliable pre-inference with reduced redundancy.

As shown in Fig. 3, PREP computes and ranks $EKL_k$ of each layer, selecting $k^*$ with the highest EKL to be pruned during full-inference.

### 3.3 MULTIMODAL TOKEN SCORE

During inference, we determine which visual tokens to prune by computing a layer-wise, token-level importance score at the EKL-selected layer $k^*$. This score fuses two complementary attention signals: intra-visual structural relevance (*visual-to-visual*, v2v) and cross-modal semantic alignment (*visual-to-text*, v2t). By combining them, we ensure that tokens critical to either visual structure or semantic information are preserved. As shown in Fig. 3, we first extract the raw multi-head attention tensor from layer $k^*$:

$$\mathbf{Attn}^{k^*} \in \mathbf{R}^{H \times (S+N+M) \times (S+N+M)}, \tag{6}$$

where $H$ is the number of attention heads, $S$ is the length of system prompts, $N$ is the number of encoded visual tokens, and $M$ is the number of text tokens. To reduce head-wise redundancy and emphasize the aggregated attention patterns, we average over all heads:

$$\overline{\mathbf{Attn}}^{k^*} = \frac{1}{H} \sum_{h=1}^{H} \mathbf{Attn}^{k^*}[h, :, :] \in \mathbf{R}^{(S+N+M) \times (S+N+M)}. \tag{7}$$

We then extract the submatrices corresponding to visual-visual and visual-text attention:

$$\mathbf{Attn}_v^{k^*} = \overline{\mathbf{Attn}}^{k^*}[S : S + N, S : S + N], \quad \mathbf{Attn}_t^{k^*} = \overline{\mathbf{Attn}}^{k^*}[S : S + N, S + N :], \quad (8)$$

where $\mathbf{Attn}_v^{k^*} \in \mathbb{R}^{N \times N}$ captures intra-visual structural interactions and $\mathbf{Attn}_t^{k^*} \in \mathbb{R}^{N \times M}$ captures visual-text semantic alignment. Then we average on the col-dimension to obtain two kinds of visual importance:

$$s_v[i] = \frac{1}{N} \sum_{j=1}^{N} \mathbf{Attn}_v^{k^*}[i, j], \quad s_t[i] = \frac{1}{M} \sum_{j=1}^{M} \mathbf{Attn}_t^{k^*}[i, j]. \quad (9)$$

Finally, we define the Multi–modal token score as the sum of visual and semantic contributions:

$$Score[i] = s_v[i] + s_t[i], \quad \mathbf{Score} \in \mathbf{R}^N. \quad (10)$$

Higher multi–modal score indicates that the $i$-th visual token is important for maintaining both visual structural integrity and cross-modal semantic alignment. During pruning, we retain the top-$m\%$ of visual tokens with the highest multi-modal token scores, ensuring that the most informative tokens are preserved.

### 3.4 THEORETICAL ANALYSIS OF REDUCED FLOPS

Following the PDrop Xing et al. (2024) approximation, the FLOPs of a single transformer layer with visual sequence length $N$ and dimension $D$ is

$$\text{FLOPs}_{\text{layer}}(N) \approx 4ND^2 + 2N^2D + 3NDC, \quad (11)$$

where $C$ is the intermediate size of the feed-forward network. As we prune at layer $k^*$ by retaining m% of the visual tokens and introduces overhead of EKL and multi–modal score, the total theoretical FLOPs reduction simplifies to, where the detail of derivation is shown in Appendix A.3:

$$\begin{aligned}
\text{Reduced FLOPs} = \sum_{k=k^*}^{K} \Big[ & 4ND^2 + 2N^2D + 3NDC \\
& - \Big( 4m \cdot ND^2 + 2(m \cdot N)^2D + 3m \cdot NDC \Big) \Big] \\
& - \Big[ k^* \cdot \big( 4rD^2 + 2r^2D + 3rDC \big) + N + HN^2 + HNM \Big].
\end{aligned} \quad (12)$$

Experiments show that the introduced overhead is only 0.02 TFLOPs, accounting for merely 0.4% of the original total FLOPs.

## 4 EXPERIMENT

### 4.1 EXPERIMENT SETTING

To assess the effectiveness of our method on image understanding tasks, we conduct experiments on four fine-grained benchmarks including MMStar Chen et al. (2024b), TextVQA Singh et al. (2019), AI2D Kembhavi et al. (2016) and Seed2-Plus Li et al. (2024), and four widely used VQA benchmarks including POPE Li et al. (2023), RealWorldQA x.ai. (2024), MME and VizWiz. At the same time,we compare PREP with recent state-of-the-art methods as SparseVLM, ToMe Bolya et al. (2022), TopV Yang et al. (2025a), FastV Chen et al. (2024a), PDrop and Minimonkey Huang et al. (2024). We verify the generalizability of PREP on InternVL3 ,LLaVA-1.5 and Qwen2.5-VL series VLMs, pruning between 6-15 layers of them. Besides, as LLaVA-1.5 gengerate fixed-size 576 visual tokens, we select group size from 32,64,144 and 192. As InternVL3 and Qwen2.5-VL set a fixed-size patch sequence length, we group visual tokens according their original size. LLaVA-1.5 employs CLIP-pretrained ViT-L as the visual tower, while InternVL3 owns dynamic high resolution encoder. All experiments are done on one NVIDIA RTX3090 with 24GB.

Table 1: Evaluation of our method on the LLaVA-1.5-7B model across nine datasets under three visual token compression levels (192, 128, and 64). The vanilla configuration uses 576 tokens and average 4.8T FLOPs. FLOPs ratio shows the ratio of pruned FLOPs to original FLOPs. Relative score is the average ratio between the score and original score across all benchmarks. Latency is measured in seconds per iteration.

| Method | Venue | MMB | MME | POPE | VizWiz | TextVQA | RWQA | AI2D | MMStar | Seed2 | Relative Score(%) | FLOPs Ratio(%) | Latency (s/it) |
|---|---|---|---|---|---|---|---|---|---|---|---|---|---|
| Original | - | 64.8 | 1864 | 86.1 | 50.0 | 58.2 | 49.0 | 52.0 | 32.9 | 38.8 | **100.0%** | 100% | 0.48 |
| | | | | | | **Retain Tokens 192** | | | | | | | | |
| ToMe | ICLR'23 | 60.5 | 1563 | 72.4 | 50.8 | 53.1 | 47.5 | 50.0 | 30.3 | 36.1 | 92.5% (↓7.5%) | 44% | 0.41 |
| FastV | ECCV'24 | 61.0 | 1605 | 64.8 | 50.9 | 52.1 | 47.9 | 50.5 | 30.5 | 36.5 | 92.1% (↓7.9%) | 46% | 0.40 |
| SparseVLM | ICML'25 | 62.5 | 1787 | 85.1 | 50.5 | 57.8 | 48.2 | 51.5 | 31.7 | 38.3 | 98.2% (↓1.8%) | 52% | 0.45 |
| PDrop | CVPR'25 | 63.3 | 1797 | 82.3 | 51.1 | 56.5 | 48.4 | 51.3 | 31.8 | 37.8 | 97.7% (↓2.3%) | **44%** | 0.42 |
| **PREP** | - | **64.8** | **1867** | **85.3** | **52.0** | **58.0** | **48.8** | **51.9** | **32.8** | **38.9** | **100.2% (↑0.2%)** | 46% | **0.39** |
| | | | | | | **Retain Tokens 128** | | | | | | | | |
| ToMe | ICLR'23 | 53.3 | 1343 | 62.8 | 50.6 | 49.1 | 44.9 | 48.0 | 28.7 | 34.2 | 85.8% (↓14.2%) | 37% | 0.37 |
| FastV | ECCV'24 | 56.1 | 1490 | 53.4 | 51.3 | 50.5 | 45.3 | 49.0 | 29.3 | 35.7 | 87.3% (↓12.7%) | 39% | 0.39 |
| SparseVLM | ICML'25 | 60.0 | 1746 | **85.0** | 51.4 | 57.0 | 45.5 | 51.0 | 31.5 | 38.0 | 96.6% (↓3.4%) | 36% | 0.42 |
| PDrop | CVPR'25 | 61.6 | 1761 | 82.3 | 51.0 | 56.6 | 46.2 | 51.2 | 32.1 | 37.9 | 96.5% (↓3.5%) | **35%** | 0.38 |
| **PREP** | - | **64.2** | **1845** | 84.9 | **51.6** | **57.5** | **47.5** | **51.4** | **32.4** | **38.6** | **99.1% (↓0.9%)** | 38% | **0.35** |
| | | | | | | **Retain Tokens 64** | | | | | | | | |
| ToMe | ICLR'23 | 43.7 | 1138 | 52.5 | 50.4 | 45.3 | 43.8 | 45.1 | 25.9 | 32.2 | 78.4% (↓21.6%) | 26% | 0.33 |
| FastV | ECCV'24 | 47.2 | 1255 | 38.2 | 51.8 | 47.8 | 42.2 | 46.3 | 26.7 | 33.1 | 79.1% (↓20.9%) | 28% | 0.34 |
| SparseVLM | ICML'25 | 56.2 | 1589 | 77.5 | 50.1 | 53.4 | 46.2 | 50.3 | 30.5 | 37.5 | 92.7% (↓7.3%) | 30% | 0.37 |
| PDrop | CVPR'25 | 58.8 | 1561 | 55.9 | 50.7 | 50.6 | 45.4 | 50.5 | 31.3 | 37.3 | 89.7%(↓10.3%) | **26%** | 0.35 |
| **PREP** | - | **63.7** | **1827** | **84.0** | **51.9** | **56.5** | **46.9** | **50.9** | **31.9** | **38.3** | **98.3% (↓1.7%)** | 29% | **0.32** |
| | | | | | | **Retain Tokens 16** | | | | | | | | |
| **PREP** | - | 63.3 | 1812 | 82.1 | 50.2 | 53.9 | 45.6 | 50.4 | 31.6 | 37.6 | **96.2% (↓3.8%)** | 27% | **0.29** |

Table 2: Performance comparison with TopV and Minimonkey on InternVL3 and Qwen2.5-VL VLMs. Latency is measured in seconds per iteration.

| Model | Method (Retained Ratio) | Venue | MMB | MME | POPE | TextVQA | OCRB | AI2D | MMStar | Seed2 | FLOPs Ratio(%) | Latency (s/it) |
|---|---|---|---|---|---|---|---|---|---|---|---|---|
| | original(100%) | - | 83.4 | 2415 | 91.1 | 81.8 | 880 | 69.7 | 85.2 | 68.2 | 100% | 0.94 |
| | TopV (50%) | CVPR'25 | 82.9 | 2407 | 89.6 | 80.4 | 825 | 66.6 | 84.5 | 67.2 | 62% | 0.87 |
| | Minimonkey(50%) | ICLR'25 | 81.7 | 2388 | 89.8 | 81.2 | 846 | 67.1 | 84.7 | 66.9 | 65% | 0.89 |
| InternVL3-8B | **PREP(50%)** | - | **83.5** | **2416** | **90.2** | **81.6** | **864** | **67.8** | **85.2** | **67.8** | **57%** | **0.84** |
| | TopV(25%) | CVPR'25 | 82.1 | 2298 | 88.2 | 78.6 | 783 | 62.4 | 83.1 | 65.3 | 46% | 0.59 |
| | Minimonkey(25%) | ICLR'25 | 81.5 | 2368 | 89.6 | 78.7 | 806 | 63.7 | 84.5 | 67.2 | 48% | 0.63 |
| | **PREP(25%)** | - | **83.1** | **2385** | **89.8** | **79.3** | **816** | **64.1** | **84.8** | **67.4** | **39%** | **0.52** |
| | original(100%) | - | 83.5 | 2305 | 86.2 | 84.9 | 864 | 81.1 | 63.9 | 70.4 | 100% | 1.12 |
| | TopV(50%) | CVPR'25 | 79.8 | 2173 | 82.4 | 81.6 | 743 | 76.2 | 61.6 | 64.2 | 64% | 0.83 |
| | Minimonkey(50%) | ICLR'25 | 80.6 | 2132 | 81.1 | 80.6 | 764 | 74.3 | 60.2 | 61.5 | 65% | 0.79 |
| Qwen2.5VL-7B | **PREP(50% )** | - | **81.7** | **2216** | **84.9** | **82.5** | **807** | **78.7** | **62.8** | **66.4** | **62%** | **0.72** |
| | TopV(25%) | CVPR'25 | 78.1 | 1973 | 78.1 | 77.3 | 711 | 71.6 | 58.3 | 62.7 | 43% | 0.68 |
| | Minimonkey(25%) | ICLR'25 | 77.6 | 2034 | 79.3 | 76.5 | 737 | 72.3 | 57.1 | 60.3 | 46% | 0.71 |
| | **PREP(25%)** | - | **80.7** | **2157** | **82.6** | **80.4** | **792** | **77.5** | **60.3** | **63.5** | **40%** | **0.63** |

## 4.2 MAIN RESULTS

Table 1 reports the performance of PREP on LLaVA-1.5-7B. We evaluate three target token budgets (192, 128, and 64) to assess compression under different levels of pruning. For the balance, we set similar computational overhead(TFLOPs) and compare both performance and latency. When reducing from 576 to 192 tokens, PREP even improves 0.2% on average accuracy, substantially lower than the drop of SparseVLM(1.8%) and PDrop (2.3%). At more aggressive pruning (16 tokens), PREP the drops only 3.8%, while other methods like FastV and ToMe retain 64 tokens and even drop more than 20%. Furthermore, we extend our approach to the advanced InternVL3 models in Table 2: when retaining only 25% of visual tokens with an average 1500 tokens per sample (far more than in LLaVA-1.5), PREP still keeps the average accuracy loss below 10%. Compared to TopV and Minimonkey on InternVL and Qwen2.5-VL, our method still achieves higher performance under the same token budget, highlighting the generalization and effectiveness of our approach. In addition, PREP achieves the lowest latency across all baselines, showing negligible pre-inference overhead and higher efficiency. Comparasion on different scales of VLMs are in Appendix A.4.

Table 3: Ablation study of EKL components under 64 tokens retained.

| Component | MME | MMBench | MMStar |
|---|---|---|---|
| Entropy | 1816 | 63.2 | 31.5 |
| KL | 1809 | 62.9 | 31.2 |
| EKL | 1827 | 63.7 | 31.9 |

Table 4: Ablation study of the $k$-th EKL score under 64 tokens retained from layer 10 to 15.

| $k$-th score | 1 | 2 | 3 | 4 | 5 | 6 |
|---|---|---|---|---|---|---|
| MME | 1768 | 1801 | 1805 | 1804 | 1819 | 1845 |
| TextVQA | 55.1 | 55.4 | 56.2 | 55.7 | 56.1 | 57.1 |
| POPE | 81.0 | 81.7 | 81.8 | 82.5 | 82.9 | 84.5 |

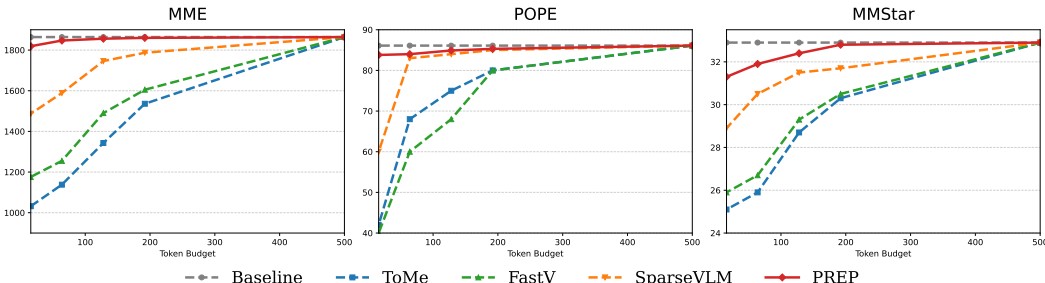

Figure 5: Performance comparison with other baselines under different tokens. The horizontal axis represents the remaining tokens to 576, 192, 128, 64 and 16, while the vertical axis means the scores.

Table 5: Counts of selected layers on MME, MMBench and SEED2.

| Layer | 6-8 | 8-10 | 10-12 | 12-14 |
|---|---|---|---|---|
| MME | 680 | 828 | 205 | 651 |
| MMBench | 2246 | 1230 | 1350 | 1864 |
| SEED2-PLUS | 780 | 501 | 820 | 176 |

Table 6: Impact of group size on performance across benchmarks.

| Group size | 32 | 64 | 144 | 192 |
|---|---|---|---|---|
| MME | 1804 | 1827 | 1806 | 1793 |
| MMBench | 64.2 | 64.5 | 63.7 | 63.2 |
| SEED2-PLUS | 31.6 | 31.9 | 31.3 | 31.1 |

Fig. 5 visualizes the performance degradation of our method compared with ToMe, FastV, and SparseVLM on POPE, MME, and MMStar under different numbers of retained visual tokens. It can be observed that even when the number of tokens is reduced to 16, our method is hardly affected by the reduction in the number of tokens on MME and POPE. Furthermore, on the MMStar dataset—which requires fine-grained perception—the magnitude of performance degradation of our method is significantly smaller than that of the other methods. We attribute this to the fact that EKL effectively identifies the layers where information fusion takes place. Combined with multi-modal token scores, PREP prevents the loss of details. These results demonstrate both the effectiveness and strong generalization of our approach.

## 4.3 ABLATION STUDY

**EKL** Table 3 compares three variants of our layer scoring: using only KL divergence, only entropy, or their combination. The results show that integrating both yields the best performance, confirming the complementarity of the two terms. Table 4 further examines the effect of selecting the $k$-th highest scoring layer, where performance consistently declines as $k$ decreases, demonstrating that EKL effectively ranks layer importance.

Table 5 shows that the majority of pruning occurs within layers 6–10, indicating that EKL is able to identify the onset of cross-modal fusion at an early stage rather than simply selecting deeper layers. This property substantially enhances the efficiency of the model.

Table 7: Performance of different variants on four benchmarks.

| | POPE | MME | TextVQA | Seed2 |
|---|---|---|---|---|
| v2t | 83.7 | 1806.3 | 56.1 | 38.0 |
| v2v | 83.5 | 1815.4 | 55.8 | 37.8 |
| ours | 84.0 | 1827.2 | 56.5 | 38.3 |

Finally, in Table 6, we investigate the impact of the number of tokens per group used in average pooling. We observe that grouping 64 tokens achieves the best performance: it preserves fine details that support reasoning while maintaining low inference overhead.

**Multi–modal token score.** Table 7 reports an ablation of multi–modal token score comparing three variants: **v2t** (using only visual-to-text attention), **v2v** (using only visual-to-visual attention), and **ours** (the full multi-modal token score that fuses v2v and v2t). Combining both signals (ours) yields the best result on all four benchmarks. For example, POPE accuracy increases from 83.9% (v2t) and 83.5% (v2v) to 84.0% (ours), and the MME score rises from 1842.3 / 1827.4 to 1856.2. Small but consistent improvements are also observed on TextVQA and Seed2-PLUS. These results show that intra-visual structure and cross-modal alignment provide complementary information for token selection, and their fusion produces more robust pruning decisions.

## 4.4 EFFICIENCY ANALYSIS

In Table 8,we evaluate the practical efficiency of our method on a single NVIDIA RTX 3090 (24GB) using full benchmarks. As our method progressively compresses visual tokens, both latency and KV cache usage are significantly reduced. For instance, decreasing the retained token count from 576 to 192 reduces latency from 0.48 s to 0.39 s, yielding a $1.23\times$ speedup, while KV cache occupancy drops nearly by half (from 100% to 56%). Further compression to 128 tokens decreases latency to 0.35 s ($1.37\times$ speedup) and KV cache usage to 44%, with minimal impact on the average performance across benchmarks (99.3%). Retaining only 64 tokens accelerates inference to 0.32 s ($1.50\times$ speedup) and reduces KV cache to 39%, whereas a further reduction to 16 tokens achieves the highest speedup of $1.66\times$, with KV cache occupancy lowered to 33%, albeit with a modest decrease in average performance (96.2%). These results demonstrate that our method effectively balances computational efficiency and model accuracy, substantially reducing memory and runtime demands while maintaining high performance on average across multiple benchmarks.

Table 8: Performance, latency, and KV cache usage comparison under different visual token configurations.

| Retain tokens | 576 | 192 | 128 | 64 | 16 |
| --- | --- | --- | --- | --- | --- |
| Performance (%) | 100 | 100 | 99.3 | 98.3 | 96.2 |
| KV Cache (%) | 100 | 56 | 44 | 39 | 33 |
| Latency (s) | 0.48 | 0.39 | 0.35 | 0.32 | 0.29 |
| Speedup ($\times$) | 1.00 | 1.23 | 1.37 | 1.50 | 1.66 |

## 4.5 CASE STUDY

As shown in Fig. 6, our method first identifies the cross-modal alignment layer via pre-inference in Stage 1, and then prunes tokens at that layer based on multi-modal token scores. The visualization highlights that our approach preserves tokens essential for answering, focusing on regions of interest.

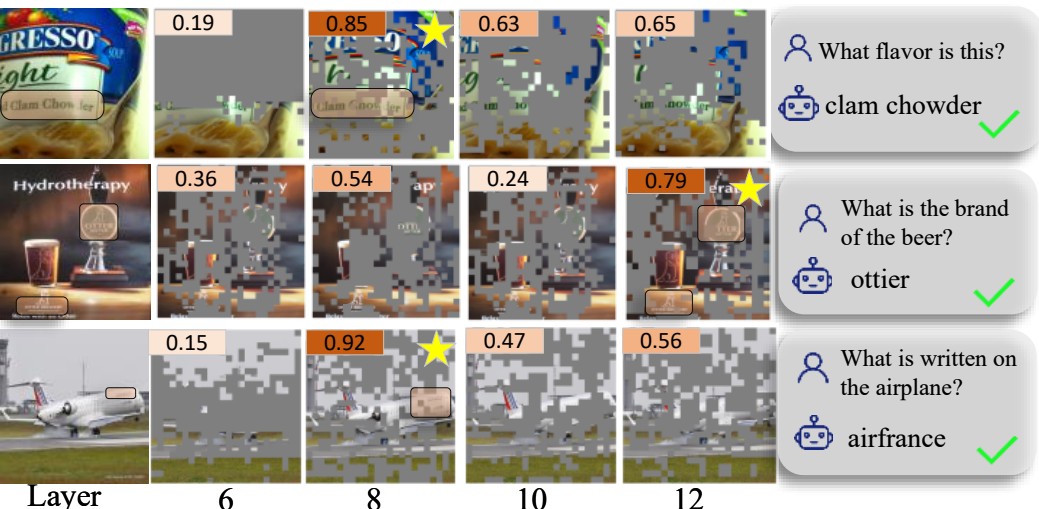

Figure 6: Visualization of our method. EKL scores are on the upper left and figures with star are the pruned layers. Orange boxes indicate regions of interest.

## 5 CONCLUSION

In this work, we introduced **PREP**, a training-free pruning framework for efficient inference in Visual-Language Models. By leveraging pooled patch-level tokens for pre-inference, PREP identifies pruning layers guided by the Information Bottleneck criterion, thereby avoiding the loss of fine-grained information that commonly arises in stepwise pruning. At the selected layer, PREP retains tokens based on multimodal importance scores, ensuring both structural integrity and semantic alignment are preserved. Extensive experiments across nine VQA benchmarks demonstrate that PREP achieves substantial efficiency gains—reducing visual tokens by up to **97%**, KV-cache usage by **67%**, and inference time by **1.66×**—while maintaining over **96%** of the original model performance. These results highlight the effectiveness of pre-inference guided pruning for high-resolution VLMs, offering a general and scalable solution toward more efficient multimodal reasoning.

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

## A  APPENDIX

### A.1  CROSS-MODAL ALIGNMENT IN OTHER VLMS.

In Fig. 7, we also observed cross-modal alignment emerging in the intermediate layers of both the InternVL and QwenVL models. This demonstrates that pretraining for next token prediction induces similar attention patterns, eliminating the need to determine distinct pruning layers for different models.

### A.2  THEORETICAL ANALYSIS OF EKL SCORE.

Let $P(i, j)$ be the joint distribution of visual tokens at adjacent layers $k - 1$ and $k$, where $i$ indexes tokens at layer $k - 1$ and $j$ at layer $k$. The marginals are:

$$\mathbf{Q}^{k-1}(i) = \sum_j P(i, j), \quad \mathbf{Q}^k(j) = \sum_i P(i, j)$$

We begin with the definition of conditional entropy:

$$H(\mathbf{Q}^k \mid \mathbf{Q}^{k-1}) = -\sum_{i,j} P(i, j) \log P(j \mid i) \tag{13}$$

$$= -\sum_{i,j} P(i, j) \log \left( \frac{P(i, j)}{\mathbf{Q}^{k-1}(i)} \right) \tag{14}$$

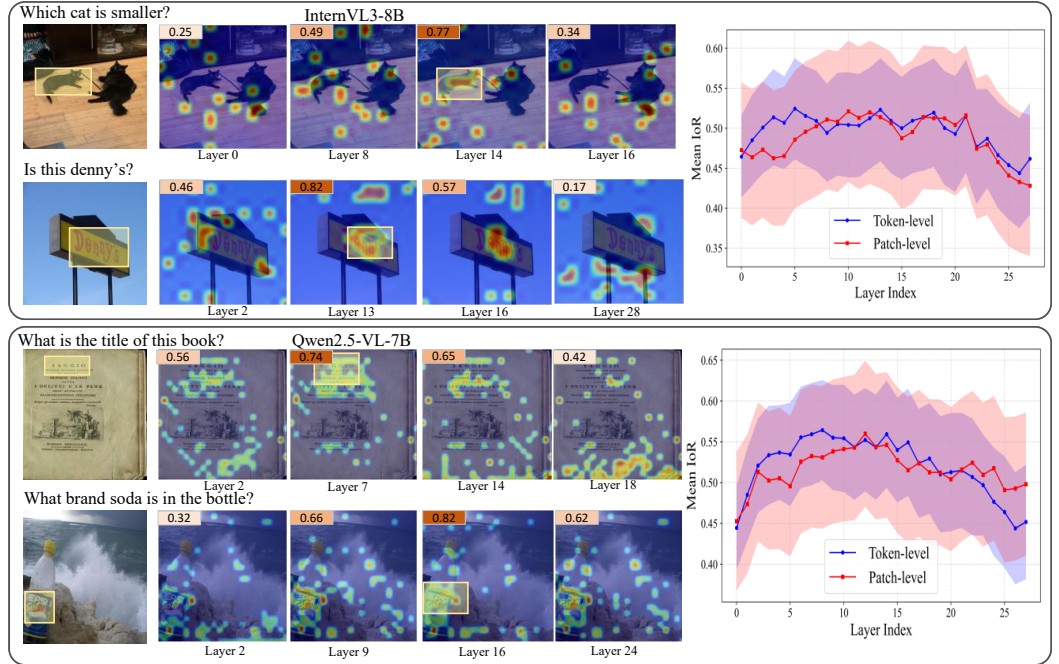

Figure 7: Attention matrices and IoR of InternVL3-8B and Qwen2.5-VL-7B across different layers, after filtering out tokens with attention weights below 70% of the maximum. EKL scores are on the upper left and yellow boxes indicate regions of interest.

Now consider the KL divergence between the marginals:

$$D_{\mathrm{KL}}(\mathbf{Q}^k \| \mathbf{Q}^{k-1}) = \sum_j \mathbf{Q}^k(j) \log \frac{\mathbf{Q}^k(j)}{\mathbf{Q}^{k-1}(j)} \tag{15}$$

$$= \sum_j \left( \sum_i P(i,j) \right) \log \frac{\mathbf{Q}^k(j)}{\mathbf{Q}^{k-1}(j)} \tag{16}$$

To establish a connection, we introduce the conditional distribution $P(j \mid i)$ and examine the relationship between $P(j \mid i)$ and $\mathbf{Q}^k(j)$.

Consider the following decomposition:

$$H(\mathbf{Q}^k \mid \mathbf{Q}^{k-1}) = -\sum_{i,j} P(i,j) \log \mathbf{Q}^k(j) + \sum_{i,j} P(i,j) \log \frac{\mathbf{Q}^k(j)}{P(j \mid i)} \tag{17}$$

$$= H_{\mathrm{cross}}(\mathbf{Q}^k, \mathbf{Q}^k) + \mathbb{E}_{i \sim \mathbf{Q}^{k-1}} \left[ D_{\mathrm{KL}}(P(\cdot \mid i) \| \mathbf{Q}^k) \right] \tag{18}$$

Note that $H_{\mathrm{cross}}(\mathbf{Q}^k, \mathbf{Q}^k) = H(\mathbf{Q}^k)$, so:

$$H(\mathbf{Q}^k \mid \mathbf{Q}^{k-1}) = H(\mathbf{Q}^k) + \mathbb{E}_{i \sim \mathbf{Q}^{k-1}} \left[ D_{\mathrm{KL}}(P(\cdot \mid i) \| \mathbf{Q}^k) \right] \tag{19}$$

From equation (19), we see that the conditional entropy equals the marginal entropy add the expected KL divergence between the conditional and marginal distributions.

Now, consider the following information-theoretic bound:

$$D_{\mathrm{KL}}(\mathbf{Q}^k \| \mathbf{Q}^{k-1}) = \sum_j \mathbf{Q}^k(j) \log \frac{\mathbf{Q}^k(j)}{\mathbf{Q}^{k-1}(j)} \tag{20}$$

$$\leq \mathbb{E}_{i \sim \mathbf{Q}^{k-1}} \left[ D_{\mathrm{KL}}(P(\cdot \mid i) \| \mathbf{Q}^{k-1}) \right] \tag{21}$$

This inequality follows from the convexity of KL divergence and an application of Jensen's inequality. Specifically:

1. **Convexity of KL divergence**: For a fixed distribution $Q$, the function $P \mapsto D_{\mathrm{KL}}(P\|Q)$ is convex in its first argument. This means that for any two distributions $P_1$ and $P_2$, and any $\lambda \in [0,1]$:

$$D_{\mathrm{KL}}(\lambda P_1 + (1-\lambda)P_2\|Q) \leq \lambda D_{\mathrm{KL}}(P_1\|Q) + (1-\lambda)D_{\mathrm{KL}}(P_2\|Q)$$

2. **Application of Jensen's inequality**: Let $P_i = P(\cdot \mid i)$ be the conditional distribution at layer $k$ given token $i$ at layer $k-1$. The marginal distribution at layer $k$ is:

$$\mathbf{Q}^k = \mathbb{E}_{i \sim \mathbf{Q}^{k-1}}[P_i]$$

By Jensen's inequality applied to the convex function $P \mapsto D_{\mathrm{KL}}(P\|\mathbf{Q}^{k-1})$, we have:

$$D_{\mathrm{KL}}\left(\mathbb{E}_{i \sim \mathbf{Q}^{k-1}}[P_i]\|\mathbf{Q}^{k-1}\right) \leq \mathbb{E}_{i \sim \mathbf{Q}^{k-1}}\left[D_{\mathrm{KL}}(P_i\|\mathbf{Q}^{k-1})\right]$$

Substituting $\mathbf{Q}^k = \mathbb{E}_{i \sim \mathbf{Q}^{k-1}}[P_i]$ and $P_i = P(\cdot \mid i)$ gives the desired inequality.

When the transformation between layers is *sufficiently smooth* and *information-preserving*, we can make the key approximation:

$$\mathbb{E}_{i \sim \mathbf{Q}^{k-1}}\left[D_{\mathrm{KL}}(P(\cdot \mid i)\|\mathbf{Q}^k)\right] \approx \mathbb{E}_{i \sim \mathbf{Q}^{k-1}}\left[D_{\mathrm{KL}}(P(\cdot \mid i)\|\mathbf{Q}^{k-1})\right] \tag{22}$$

This approximation holds when $\mathbf{Q}^k$ and $\mathbf{Q}^{k-1}$ are similar, which is reasonable for adjacent layers in a well-trained neural network.

Substituting approximation (22) into equation (19):

$$H(\mathbf{Q}^k \mid \mathbf{Q}^{k-1}) \approx H(\mathbf{Q}^k) + \mathbb{E}_{i \sim \mathbf{Q}^{k-1}}\left[D_{\mathrm{KL}}(P(\cdot \mid i)\|\mathbf{Q}^{k-1})\right] \tag{23}$$

$$\geq H(\mathbf{Q}^k) + D_{\mathrm{KL}}(\mathbf{Q}^k\|\mathbf{Q}^{k-1}) \tag{24}$$

Finally, by substituting this inequality into (3), and considering that when IoR is similar, $H(\mathbf{Q}^k \mid Y)$ remains relatively constant, we can bound (2) as:

$$\mathcal{L}_{IB} \geq H(\mathbf{Q}^k) + D_{\mathrm{KL}}(\mathbf{Q}^k\|\mathbf{Q}^{k-1}) + \text{const}, \tag{25}$$

which is the EKL computation method. Therefore, EKL provides a computationally tractable surrogate for analyzing information flow through the network layers and identifying cross-modal alignment.

### A.3 THEORETICAL ANALYSIS OF REDUCED FLOPs.

We prune visual tokens at layer $k^*$, retaining only the top $m\%$ of $N$ visual tokens. Below we compute FLOPs explicitly in terms of model dimensions.

**Transformer layer FLOPs.** For a Transformer layer with visual sequence length $N$, hidden dimension $D$, intermediate size of feed-forward network $C$, the approximate FLOPs is:

$$\mathrm{FLOPs_{layer}}(N) = 4ND^2 + 2N^2D + 3NDC. \tag{26}$$

**Pre-inference FLOPs.** Before pruning, we partition N visual tokens into r groups and average them($N = rL$), which takes $rL$ FLOPs. Then, we use them to pre-inference up to layer $k^*$, which takes FLOPs:

$$\mathrm{FLOPs_{pre\text{-}inference}} = k^* \cdot \mathrm{FLOPs_{layer}}(r) + N. \tag{27}$$

Then, computing EKL requires entropy and KL divergence over $r + M$ tokens:

$$\mathrm{FLOPs_{EKL}} \sim O((r + M)D), \tag{28}$$

**Multi-modal token score computation FLOPs.** At layer $k^*$, computing multi-modal token score involves:

1. Averaging attention over $H$ heads for v2v: $\mathrm{FLOPs_{v2v}} = H \cdot N^2$,

2. Averaging attention over $H$ heads for v2t: $\mathrm{FLOPs_{v2t}} = H \cdot (N \cdot M)$.

Thus the total multi-modal token score overhead is

$$\mathrm{FLOPs_{multi\text{-}modal\ token\ score}} \approx HN^2 + HNM. \tag{29}$$

**Inference FLOPs after pruning.** After pruning $100 - m\%$ of visual tokens, the sequence length becomes

$$N_{\text{pruned}} = m \cdot N. \tag{30}$$

The FLOPs per layer in the upper layers $k^*, \ldots, K$ are

$$\text{FLOPs}_{\text{layer}}(N_{\text{pruned}}) = 4N_{\text{pruned}}D^2 + 2N_{\text{pruned}}^2 D + 3N_{\text{pruned}}DC. \tag{31}$$

**Explicit expression.** Substituting $N_{\text{full}} = N + M$ and $N_{\text{pruned}} = m \cdot N + M$, and using the standard transformer FLOPs formula $\text{FLOPs}_{\text{layer}}(N) = 4ND^2 + 2N^2D + 3ND^2/H$, the reduced FLOPs can be written explicitly as

$$
\begin{aligned}
\text{Reduced FLOPs} = \sum_{k=k^*}^{K} &\Big[ 4ND^2 + 2N^2D + 3NDC \\
&- \Big( 4m \cdot ND^2 + 2(m \cdot N)^2 D + 3(m \cdot N)DC \Big) \Big] \\
&- \Big[ k^* \cdot \big( 4rD^2 + 2r^2D + 3rDC \big) + N + HN^2 + HNM \Big].
\end{aligned} \tag{32}
$$

**Intuition.** The first term captures the main savings from pruning the visual sequence in upper layers. The second term accounts for pre-inference, EKL and multi-modal token score computation.

A.4 COMPARISON ON DIFFERENT PARAMETER SCALES

In Tab. 9, we conducted a performance comparison between the InternVL3-2B and Qwen2.5-VL-3B models and found that PREP consistently achieved favorable results across different parameter scales.

Table 9: Performance comparison with TopV and Minimonkey on InternVL3-2B and Qwen2.5-VL-3B. Latency is measured in seconds per iteration.

| Model | Method(Retained Ratio) | Venue | MMB | MME | POPE | TextVQA | OCRBench | AI2D | MMStar | Seed2 | FLOPs Ratio(%) | Latency (s/it) |
|---|---|---|---|---|---|---|---|---|---|---|---|---|
| InternVL3-2B | original(100%) | - | 80.3 | 2180 | 89.6 | 77.0 | 835 | 78.7 | 78.6 | 64.6 | 100% | 0.65 |
| | TopV(50%) | CVPR'25 | 79.4 | 2076 | 88.4 | 75.2 | 795 | 77.4 | 76.8 | 62.5 | 59% | 0.52 |
| | Minimonkey(50%) | ICLR'25 | 79.7 | 2096 | 88.7 | 75.5 | 802 | 77.8 | 77.0 | 62.9 | 65% | 0.55 |
| | **PREP(50%)** | - | **80.2** | **2195** | **90.0** | **76.8** | **822** | **78.3** | **78.0** | **63.8** | **52%** | **0.47** |
| | TopV(25%) | CVPR'25 | 78.5 | 2042 | 87.6 | 72.5 | 705 | 76.2 | 74.5 | 62.2 | 46% | 0.39 |
| | Minimonkey(25%) | ICLR'25 | 78.7 | 2068 | 87.9 | 72.8 | 721 | 76.4 | 74.8 | 62.4 | 48% | 0.42 |
| | **PREP(25%)** | - | **80.3** | **2171** | **89.8** | **73.0** | **746** | **77.6** | **77.8** | **63.4** | **36%** | **0.33** |
| Qwen2.5VL-3B | original(100%) | - | 79.1 | 2157 | 83.6 | 79.3 | 797 | 81.6 | 55.9 | 67.6 | 100% | 0.72 |
| | TopV(50%) | CVPR'25 | 73.5 | 1895 | 82.1 | 75.8 | 698 | 74.8 | 50.2 | 60.8 | 62% | 0.59 |
| | Minimonkey(50%) | ICLR'25 | 74.2 | 1912 | 82.5 | 76.2 | 705 | 75.3 | 50.6 | 61.3 | 66 % | 0.63 |
| | **PREP(50%)** | - | **76.7** | **1971** | **85.1** | **77.9** | **721** | **76.9** | **52.2** | **63.2** | **61%** | **0.51** |
| | TopV(25%) | CVPR'25 | 71.8 | 1823 | 80.5 | 73.2 | 675 | 72.1 | 48.7 | 59.2 | 44% | 0.44 |
| | Minimonkey(25%) | ICLR'25 | 72.5 | 1845 | 81.0 | 73.8 | 682 | 72.8 | 49.1 | 59.8 | 46% | 0.49 |
| | **PREP(25%)** | - | **74.2** | **1864** | **83.7** | **75.1** | **703** | **74.8** | **50.7** | **61.7** | **42%** | **0.38** |

A.5 THE USE OF LARGE LANGUAGE MODELS (LLMS)

In this work, we employed ChatGPT as an auxiliary writing tool to improve the clarity and readability of the manuscript. Specifically, ChatGPT was used to refine the language of the *Abstract*, *Introduction*, and *Conclusion* sections. No part of the technical content, experimental design, or results was generated or modified by LLMs.

