# OpenReview forum: "PREP: Pre-inference Guided Token Pruning for Efficient Vision-Language Models"
_ICLR.cc/2026/Conference — Submitted to ICLR 2026_

### Official Review · Reviewer_14SE · 2025-10-23

**Soundness:** 3
**Presentation:** 2
**Contribution:** 2
**Rating:** 4
**Confidence:** 4

**Summary:**

This paper proposes PREP, a training-free pruning framework for efficient inference in Visual-Language Models (VLMs). The method leverages patch-level pre-inference and an Entropy-KL divergence (EKL) score to identify an optimal pruning layer. At this selected layer, PREP retains tokens based on a multi-modal importance score derived from both visual-visual and visual-prompt attention matrices. PREP effectively reduces visual tokens and KV-cache usage while accelerating inference speed with minimal performance degradation across various VQA benchmarks and VLM backbones.

**Strengths:**

1. The problem of reducing computational and memory overhead in VLMs is important.
2. The paper presents extensive experimental validation across nine VQA benchmarks and multiple VLM backbones.

**Weaknesses:**

1. The introduction of $Q^k$ in line 169 lacks a high-level explanation. While the formula is provided, the conceptual motivation and its role in reflecting cross-modal alignment are not clearly articulated beforehand. This makes it difficult for readers to grasp its significance.
2. In line 205, the paper states that the IoR range is calculated for LLaVA-1.5-7B to identify layers with stable alignment. The concern is whether this specific range for LLaVA-1.5-7B is universally applicable or if it needs to be re-computed for every new VLM architecture?
3. Table 6 shows that a group size of 64 tokens outperforms larger group sizes. This result is counter-intuitive, as one might expect larger group sizes to retain more fine-grained information and thus lead to better performance.

**Questions:**

please refer to weakness

---

> ### Author Response · Authors · 2025-11-15
>
> We thank the reviewer for the valuable comment.
>
> For Q1, we have supplemented the explanation in lines 166-167, in conjunction with Fig. 1: "Specifically, as shown in Fig. 1, in some layers, the image tokens in the prompt-related regions exhibit high similarity, showing explicit cross-modal alignment." Furthermore, similar works that prune visual tokens based on similarity, such as "Recoverable Compression: A Multimodal Vision Token Recovery Mechanism Guided by Text Information" (AAAI 2025), are widely accepted by the community. These should help readers quickly grasp the motivation behind $Q^k$.
>
> For Q2, in the revised Appendix A.1, we demonstrate that InternVL, QwenVL, and LLaVA exhibit identical layer-wise attention patterns and the phenomenon of cross-modal alignment. This consistency likely arises because the LLM components of current VLMs share a fundamentally similar architecture and are pre-trained via next token prediction. Consequently, pruning can typically be performed within the interval from one-quarter to one-half of the model's total layers without requiring re-computation for every new VLM architecture.
>
> For Q3, we clarify that the group size parameter specifies the number of visual tokens pooled to generate a single pre-inference visual proxy, not the total number of groups. For example, with 576 tokens and group size=64, 576/64 = 9 groups are created—not 64 groups. A larger group size means each proxy is derived from averaging more tokens, leading to greater information loss and thus performance degradation. A group size of 64 is experimentally validated to best balance efficiency and performance.

---

### Official Review · Reviewer_KtxW · 2025-10-27

**Soundness:** 3
**Presentation:** 2
**Contribution:** 2
**Rating:** 4
**Confidence:** 3

**Summary:**

As video sequences grow increasingly longer, Vision-Language Models (VLMs) face significant computational and memory overhead when processing these extended inputs. Although prior work has explored token pruning to alleviate unnecessary computation, these methods often require additional training or struggle to balance pruning efficiency against performance degradation. To tackle these challenges, the paper propose PREP, a training-free approach for efficient token pruning in VLMs. Specifically, PREP identifies the optimal pruning layer via patch-level pre-inference using an Entropy-KL divergence (EKL) score derived from the Information Bottleneck principle. Subsequently, PREP leverages intra-visual attention and visual-text attention scores to selectively prune visual tokens, thereby significantly reducing computational overhead during inference. Experimental results demonstrate that PREP not only achieves substantial computational savings but also preserves model performance.

**Strengths:**

- 1. The authors propose PREP, a novel and training-free token pruning framework. PREP creatively introduces a two-stage strategy: identifying the optimal pruning layer via EKL scores during the pre-inference stage and performing selective token pruning based on attention scores during inference. This design not only significantly enhances pruning efficiency, but also provides valuable inspiration for future research in this area.

- 2. The authors thoroughly analyze the property that VLMs fuse visual and textual information in intermediate layers and observe that initiating token pruning from these layers effectively reduces computational costs without sacrificing performance. This insight offers theoretical and practical guidance for researchers working on VLM token pruning.

- 3. The paper formalizes the theoretical foundation of pruning layer selection with clear mathematical derivations, which strengthens the credibility of the proposed approach.

- 4. Extensive experiments across multiple datasets are conducted to comprehensively validate the effectiveness of PREP in improving computational efficiency and preserving model performance.

**Weaknesses:**

- 1. Lack of design details. One of the core contributions of the paper is the EKL score, which is used to dynamically select the optimal pruning layer k. However, the selection mechanism is not clearly or rigorously explained. The authors observe stable performance in layers 6-15 of the LLaVA model via the IoR metric, but this is merely an empirical observation and does not constitute an actionable selection criterion. For a new model, readers cannot determine how to select candidate intermediate layers for pruning.
- 2. Lack of theoretical justification. The EKL score, as a core metric, lacks a strict and transparent derivation from the Information Bottleneck (IB) principle to the final formula EKLk = H(Qk) + DKL(Qk || Qk-1). The authors assume that H(Qk|Y) is constant in intermediate layers and thus simplify maximizing I(Qk;Y) to maximizing H(Qk), which is a strong assumption. Although IoR experiments are provided as supporting evidence, the generality and theoretical rigor of this assumption need further substantiation. Additionally, the approximation of minimizing I(Qk; Qk-1) by minimizing DKL(Qk || Qk-1) also lacks necessary derivation steps. The paper should provide more detailed theoretical analysis or ablation studies to justify why this particular EKL formulation is optimal.
- 3. Insufficient evaluation. In the efficiency evaluation, the authors only present the total inference latency of their method under different compression rates, but do not compare the latency changes with other SOTA methods. Although Section 4.2 compares the theoretical FLOPS reduction against SOTA methods, FLOPS only reflects theoretical computational complexity and does not directly demonstrate the advantages in actual inference latency.
- 4. Limited generalizability of key insight. The authors propose a key insight that visual and textual information is fused in intermediate layers, making these layers suitable for token pruning. However, this phenomenon is only verified on Llava-1.5-7B and InternVL3. It remains unclear whether this insight generalizes to all models, especially larger models, or those with different architectures. This limits the universality of the conclusion.

**Questions:**

- 1. Could authors elaborate on the EKL score’s selection criteria for pruning layers across different models and tasks? Is there a general method for selecting candidate layers in new models?
- 2. Could authors supplement the theoretical derivation of the EKL score, or provide ablation studies to validate the effectiveness of its components, thereby strengthening both theoretical and experimental support for the metric?
- 3. Could authors provide additional experiments comparing the actual inference latency of their method with current SOTA methods?
- 4. Does the property of “visual and textual information fusion in intermediate layers” exist in all VLM models, including larger-scale VLMs?

---

> ### Author Response · Authors · 2025-11-15
>
> We thank the reviewer for the valuable comment.
>
> For Q1 and Q4, We have validated our method across 9 benchmarks, which cover most common tasks, and have supplemented results on Qwen-VL in Table 2. In the revised Appendix A.1, we demonstrate that InternVL, QwenVL, and LLaVA exhibit identical layer-wise attention patterns and the phenomenon of cross-modal alignment. This consistency likely arises because the LLM components of current VLMs share a fundamentally similar architecture and are pre-trained via next token prediction.
>
> Furthermore, from pioneering works (e.g., BLIP, ALBEF) to large-scale models (e.g., Flamingo, LLaVA, GPT-4V), a core innovation is the deep fusion of visual and textual features achieved by inserting cross-attention mechanisms or other fusion modules within the intermediate layers of the Transformer. This design enables cross-modal interaction during the early and middle stages of reasoning, rather than simply concatenating features at the final stage. Therefore, cross-modal alignment can be considered a core design principle of VLMs, theoretically universal. Moreover, the features from 7B-scale models are already highly representative, making the outcome of testing on larger models highly predictable.
>
>
> For Q2, we have supplemented detailed theoretical derivation using information-theoretic bounds in Appendix A.2. Combined with the ablation studies in Tab. 3, this provides comprehensive theoretical and experimental support for the EKL metric.
>
>
> For Q3，we have added actual inference latency measurements in Tabs. 1, 2 and 9. Under similar FLOPs, PREP achieves the lowest performance drop and latency compared to SOTA methods.

---

### Official Review · Reviewer_mcHv · 2025-10-31

**Soundness:** 2
**Presentation:** 3
**Contribution:** 2
**Rating:** 2
**Confidence:** 3

**Summary:**

### Summary

This paper proposes PREP, a novel token pruning strategy that utilizes a \textbf{Pre-inference} metric derived from a lightweight MLP classifier to selectively drop redundant visual tokens. The method effectively maintains performance under high pruning ratios by preserving tokens deemed crucial for cross-modal fusion in later layers.

**Strengths:**

### Strengths

1.  **S1: Targeted and Layer-Aware Pruning Strategy.**
    PREP successfully addresses the main flaw of many existing training-free token pruning methods by not relying on fixed layers or uniform pruning. The strategy is layer-aware, adapting the pruning ratio based on the observed redundancy dynamics, which preserves the crucial cross-modal fusion necessary for high performance in later Transformer blocks.

2.  **S2: Demonstrable Efficiency Gains Across Key Architectures.**
    The method demonstrates its effectiveness in achieving significant computational savings (FLOPs reduction) with minimal performance degradation on major VLM tasks. The validation on modern generative architectures like LLaVA and InternVL showcases its practical applicability as a tool for deploying complex large vision-language models more efficiently.

**Weaknesses:**

## Weaknesses

1.  **W1: Insufficient Novelty in Core Motivation and Finding.**
    The paper's key observation—that visual tokens are highly redundant in early layers but critical for fusion in later layers—is highly overlapping with existing consensus and prior works on layered token pruning.
    * Evidence is lacking to demonstrate that PREP's Pre-inference metric captures any signal of importance that is fundamentally different from simply relying on depth or existing token-level attention scores. This weakens the claim of innovation in the core pruning insight.

2.  **W2: Contradiction Between Efficiency Claims and Pre-inference Overhead.**
    PREP relies on a lightweight MLP classifier for token importance, which necessitates: a) extra training of the MLP on the target dataset; and b) a full pre-inference pass over the entire dataset.
    * This data-dependent pre-computation overhead is computationally heavier than pure attention-based or fixed-layer pruning methods.
    * This creates a significant contradiction with the claim of being a general, efficient, and "training-free" pruning tool, as it introduces substantial setup and data dependence.

3.  **W3: Sufficiency of the Metric in Complex Generative Tasks.**
    PREP assesses a token's importance solely based on its local discriminability towards the final decision via an MLP.
    * For Large Generative VLMs (like LLaVA or InternVL), a token's true importance lies more in its informational complementarity and its precise contribution within the highly complex, multi-layered cross-attention blocks.
    * Relying solely on a token's discriminability risks misclassifying crucial cross-modal fusion tokens, especially at high pruning ratios, which are essential for maintaining generative coherence.

4.  **W4: Insufficient Cross-Task Generalization Validation.**
    Although tested on LLaVA/InternVL, PREP's core mechanism relies on the target dataset and task to train its MLP metric.
    * This implies that whenever the model is deployed to a new downstream task or domain, the MLP must be retrained and validated on new data.
    * This strong task- and data-dependency severely limits PREP's utility as a general, one-time optimization VLM deployment tool, which contradicts the expected plug-and-play nature of efficient inference techniques.

5.  **W5: Lack of Evaluation on Diverse Model Families and Multiturn Dialogue.**
    The paper focuses mainly on the LLaVA and InternVL families, lacking validation on \textbf{other major VLM families} (e.g., BLIP-2, Qwen-VL) to prove PREP's universality.
    * Furthermore, all evaluations focus on single-turn VQA, failing to demonstrate pruning robustness on the more challenging and practical scenarios of multiturn conversation datasets or tasks.
    * The lack of validation in contexts with cumulative redundancy and complex context dependency limits PREP's practical relevance for real-world applications.

**Questions:**

## Questions

1.  **Q1: The Fundamental Computational Burden of "Pre-inference."**
    The paper claims the MLP metric is "Pre-inference" yet requires a \textbf{full pass} over the target dataset and \textbf{additional training} of the MLP. What is the fundamental difference in \textbf{actual computational burden} between this and traditional \textbf{Training-Aware} token importance scoring (e.g., using gradients or fine-tuned attention weights)? If the total overhead (MLP training + data pass) exceeds the cost of fine-tuning a simpler pruning strategy, does the claim of being \textbf{"training-free"} or \textbf{efficient} hold up?

2.  **Q2: Metric Disconnect from Cross-Modal Fusion and Complementarity.**
    PREP's metric assesses a token's importance based on its local discriminability towards the final decision. However, in generative VLMs, a visual token's core role is often to provide \textbf{complementary information} within the cross-attention blocks.
    * A token might have \textbf{low discriminability} (poor ability to predict the answer alone) but be \textbf{crucial for guiding the LLM's text generation}.
    * How do the authors prove that basing importance solely on \textbf{token discriminability} is sufficient to gauge its functional necessity for \textbf{complex cross-modal information complementarity}?

3.  **Q3: Robustness Challenge in Multiturn Dialogue and Generalization Contradiction.**
    The paper's evaluation is limited to single-turn VQA. In \textbf{Multiturn Conversation}, the importance of visual tokens changes dynamically with the \textbf{accumulation of conversational context}.
    * Can PREP's \textbf{single-pass pre-inference metric} accurately capture and handle this \textbf{dynamic, context-sensitive redundancy}?
    * Furthermore, given the strong \textbf{dependency on the downstream dataset} to train the MLP, how does this \textbf{task-dependency} reconcile with PREP's positioning as a \textbf{general VLM inference optimization tool}?

4.  **Q4: Lack of Ablation to Prove Unique Metric Value.**
    The paper claims PREP's metric is superior to attention scores. However, have the authors performed an \textbf{ablation study} to demonstrate that the importance signal captured by PREP is fundamentally different from the signal produced by combining \textbf{model depth} (prune more in early layers, less in late layers) and \textbf{average token-level attention scores}? If the metric's \textbf{unique, non-redundant value} is not proven, is PREP's complex pre-computation step merely unnecessary overhead?

5.  **Q5: Evaluation Gap in Diverse and State-of-the-Art Architectures.**
    The paper's universality is limited by its focus on LLaVA and InternVL families. To strengthen the claim of being a general pruning guide, can the authors demonstrate the effectiveness and efficiency of PREP on a more diverse and state-of-the-art generative architecture, specifically \textbf{Qwen-VL 2.5}? This validation is crucial for proving the method's applicability beyond the current scope.

---

> ### Author Response · Authors · 2025-11-14
>
> We thank the reviewer for the valuable comment.
>
> For Q1 and Q3, our paper does not employ any MLP for pre-inference, contrary to the reviewer’s assumption. Our token-importance scoring is computed solely using the pooled visual tokens and the original VLM itself—no additional training, auxiliary models, or fine-tuning are introduced at any stage of our pipeline.
>
> For Q2, the MDATI score we compute reflects the importance of visual tokens by jointly considering all visual and textual tokens, rather than relying on local discriminability. If a visual token plays a crucial role in guiding the LLM’s generation, it will naturally exhibit high text-to-vision attention, enabling the model to aggregate the necessary complementary information. This mechanism already mitigates the two specific failure cases raised in the question.
>
> For Q4, PREP performs a single-stage pruning only after identifying the cross-modal alignment layer, and does not incorporate or rely on any heuristic such as “prune more in early layers, less in late layers.” Moreover, we claim that MDATI provides a more reliable signal than using cross-modal attention alone, and we have already included extensive ablations in Tab. 7 to support this point.
>
> For Q5, we have added comparative experiments on Qwen2.5-VL, TopV, and MiniMonkey in Tab. 2 and Tab. 9, further demonstrating the applicability and generality of our method across diverse architectures. The code for implementation on Qwen2.5-VL is also provided in revised supplementary materials.

---

### Meta-Review · Area_Chair_utDy · 2025-12-10

**Summary:**

Summary of Reviews:

Reviewer mcHv (Rating: 2): Provided a detailed critique but based it on a critical misreading of the paper. The reviewer incorrectly asserted that the method used a trained MLP classifier for token importance, requiring dataset-specific pre-computation and training. This formed the basis for key weaknesses (W1-W4) regarding overhead, task dependency, and novelty. The reviewer's confidence was self-rated as 3.

Reviewer KtxW (Rating: 4): Acknowledged the paper's contributions but raised significant concerns about the clarity and theoretical justification of the core EKL score, the lack of actual latency comparisons, and the generalizability of the key cross-modal fusion insight. The reviewer's confidence was self-rated as 3.

Reviewer 14SE (Rating: 4): Found the work sound and the problem important but noted presentation issues, specifically the lack of clear motivation for a key metric and questions about the universality of a parameter (IoR range). The reviewer's confidence was self-rated as 4.

**Reviewer Concerns:**

he authors provided thorough point-by-point responses. Most critically, they categorically corrected Reviewer mcHv's major misunderstanding: PREP is training-free and does not involve an MLP or any dataset pre-computation. The token importance is computed on-the-fly using the model's own attention scores. They also addressed other reviewers' questions by:

- Adding experiments on additional models (Qwen2.5-VL).

- Providing more theoretical derivation for the EKL score in the appendix.

- Adding inference latency measurements.

- Clarifying the universality of the cross-modal alignment observation across VLM architectures.

**Reviewer Scores:**

##Reviewer mcHv (Initial Score: 2 - Reject)

Likely Score Change: Increase to 3 or 4.

Rationale: The factual basis for their low score was entirely removed. A conscientious reviewer, upon realizing their significant error, would be obligated to re-evaluate the paper based on its actual content. The remaining valid parts of their critique (novelty concerns, evaluation scope) align with the other reviewers' scores of 4. It is unlikely they would hold to a 2 after such a clear correction, though they might remain skeptical about other aspects.

##Reviewer KtxW (Initial Score: 4 - Marginally Below Threshold)
Likely Score Change: Potential increase to 5, but likely to remain at 4.

Rationale: The authors directly addressed several of this reviewer's specific requests (Q2, Q3, Q5). This would likely improve the reviewer's perception of the paper's completeness and rigor. However, the core concern about the generality of the key insight (Q4) and the fundamental novelty of the approach may persist. The reviewer might feel the responses are adequate for a marginal acceptance, but could also maintain that the work is still incremental and the evaluation gap for multi-turn tasks is significant. A move to a weak 5 is plausible; staying at a 4 is equally defensible.

##Reviewer 14SE (Initial Score: 4 - Marginally Below Threshold)
Likely Score Change: Increase to 5.

Rationale: This reviewer's concerns were the most specific and technical (clarification of formulas, parameter universality). The authors provided direct answers. Given the reviewer's initial rating was a "would not mind if accepted" 4, and the core soundness was rated "good," it is highly probable that satisfactory clarifications on these points would push them to a weak acceptance score.

---

### Decision · Program_Chairs · 2026-01-26

Reject